# Natural Inspired Carboxymethyl Cellulose (CMC) Doped with Ammonium Carbonate (AC) as Biopolymer Electrolyte

**DOI:** 10.3390/polym12112487

**Published:** 2020-10-26

**Authors:** Mohd Ibnu Haikal Ahmad Sohaimy, Mohd Ikmar Nizam Mohamad Isa

**Affiliations:** 1Energy Storage Research, Frontier Research Materials Group, Advanced Materials Team, Ionic & Kinetic Materials Research Laboratory (IKMaR), Faculty of Science & Technology, Universiti Sains Islam Malaysia, Nilai 71800, Malaysia; ibnuhyqal@gmail.com; 2Advanced Nano Materials, Advanced Materials Team, Ionic State Analysis (ISA) Laboratory, Faculty of Science & Marine Environment, Universiti Malaysia Terengganu, Kuala Nerus 21030, Malaysia

**Keywords:** biopolymer, carboxymethyl cellulose, solid polymer electrolyte, ionic transport

## Abstract

Green and safer materials in energy storage technology are important right now due to increased consumption. In this study, a biopolymer electrolyte inspired from natural materials was developed by using carboxymethyl cellulose (CMC) as the core material and doped with varied ammonium carbonate (AC) composition. X-ray diffraction (XRD) shows the prepared CMC-AC electrolyte films exhibited low crystallinity content, *X_c_* (~30%) for sample AC7. A specific wavenumber range between 900–1200 cm^−1^ and 1500–1800 cm^−1^ was emphasized in Fourier transform infrared (FTIR) testing, as this is the most probable interaction to occur. The highest ionic conductivity, *σ* of the electrolyte system achieved was 7.71 × 10^−6^ Scm^−1^ and appeared greatly dependent on ionic mobility, *µ* and diffusion coefficient, *D*. The number of mobile ions, *η,* increased up to the highest conducting sample (AC7) but it became less prominent at higher AC composition. The transference measurement, *t_ion_* showed that the electrolyte system was predominantly ionic with sample AC7 having the highest value (*t_ion_* = 0.98). Further assessment also proved that the H^+^ ion was the main conducting species in the CMC-AC electrolyte system, which presumably was due to protonation of ammonium salt onto the complexes site and contributed to the overall ionic conductivity enhancement.

## 1. Introduction

Apart from increased energy and power density of energy storage, one of the remaining challenge in advancement of energy storage technologies such as portable electronic devices, smart grids and electric vehicles is to lower the production cost as well as to reduce the environmental effect [1,2,3]. Thus, one of the steps in order to achieve this is to transform the electrolyte used in energy storage. The electrolyte is a medium that facilitates the movement of ions across the electrode in order to complete the circuit and usually the electrolyte is in liquid form since ionic transport is easier in liquid phase. However, the development of solid polymer electrolyte (SPE) film has garnered much attention as it offers outstanding advantages compared to other form of electrolytes [3,4,5]. Several types of polymer were used to investigate the possibilities for energy storage material such as polyethylene oxide (PEO), polyvinyl chloride (PVC), poly(methyl methacrylate) (PMMA), polyvinylidene fluoride (PVDF) and polyacrylonitrile (PAN) [4,6,7,8]. Alternatively, polymers originated from natural sources (biopolymer) such as starch, chitosan, gelatin and agar [9,10,11,12] are also becoming an attractive substitute for synthetic polymers partly due to superior mechanical, electrical properties and for being cheaper [9]. Equally, cellulose in particular is another good choice since it is literally found everywhere but cellulose in its original form is hard to utilize due to its inability to dissolved and shaped. However, derivatives of the base polymer can overcome this problem.

Currently several cellulose derivatives biopolymer can be found commercially such as methyl cellulose (MC), hydroxyethyl cellulose (HEC) and carboxymethyl cellulose (CMC) [13,14,15]. CMC in particular is a good SPE biopolymer candidate. This is attributable to its good film-forming abilities due to the presence of a hydrophilic carboxyl group (–CH_2_COONa) which allows the biopolymer to dissolve in cheap solvent (water) [16]. On top of that, CMC has a naturally high degree of amorphous phase [17]. These allow easier transport of conducting ions like lithium (Li^+^) and proton (H^+^). The major flaw with SPE is the low ionic conductivity due to inherent solid properties of the polymer itself, which impede ionic mobility especially at room temperature [18]. For a biopolymer electrolyte to be utilized in commercialized energy storage technology, the ionic conductivity needs to be enhanced to at least a minimum value of ~10^−4^ Scm^−1^, and one of the most prominent methods for ionic conductivity enhancement is by introducing the ionic dopant into the polymer electrolyte. Ammonium salts are the dopant materials focused upon in this research. Ammonium nitrate (A–N) and ammonium fluoride (A–F) are some examples of ammonium salts [17,19] and are the most common proton donors. Deprotonated of H^+^ from the ammonium ion group (NH_4_^+^) of ammonium salts helps to improve ionic conductivity of the electrolyte system. Thus in theory, ammonium carbonate (AC) which have two group of ammonium ions could inject more H^+^ into the system.

The ionic conductivity is the most vital parameters in order to determine the viability of the electrolyte system. Nevertheless, the value calculated only gives a general overview of the electrolyte performance. Fundamentally, the ionic conductivity of the electrolyte is the product of ionic density, *η*, ionic mobility, *µ* and the elementary charge, *e*. Thus, it can be beneficial to acquire a detailed transport number for the electrolyte system as it can help to evaluate and later suggest an appropriate technique to improve the ionic conductivity of the electrolyte system. Several methods have been employed previously by researchers to calculate the transport number [20,21]. The commonly used one is by using the Fourier transform infrared (FTIR) deconvolution approach, which is also the method chosen for this study [22,23,24]. Apart from that, a high cation transference number is also crucial in electrolyte system as it can reduce charge concentration gradient and subsequently produce higher power density when assembled in a battery [25].

This paper aims to develop SPE from a biopolymer, which is CMC doped with ammonium carbonate (AC) as the ionic source, then to find the ionic conductivity and the transport parameters contributing towards the ionic conductivity behavior in CMC-AC biopolymer films. Ionic conductivity was measured using electrical impedance spectroscopy (EIS). The X-ray diffraction (XRD) and Fourier transform infrared (FTIR) spectrum of the electrolyte was de-convoluted to quantitatively determine the crystallinity content and the transport number (number of mobile ions, *η* mobility of mobile ions, *µ* and diffusion coefficient, *D*), respectively. A direct current (dc) polarization technique was utilized to confirm the ionic transference number of CMC-AC biopolymer films. From these analyses, the correlation between the results towards the ionic conductivity behavior can be ascertained.

## 2. Materials and Methods

### 2.1. Preparation of Carboxymethyl Cellulose-Ammonium Carbonate (CMC-AC) Electrolytes Films

Carboxymethyl cellulose (CMC) (Across, NJ, USA) were dissolved in 100 mL of distilled water until homogenous before adding (1–11 wt.%) of ammonium carbonate salts (AC) (Merck, Darmstadt, Germany) which is calculated using Equation (1), where the *x* represents the weight of added salt and *y* the weight of CMC. The compositions of the CMC-AC biopolymer films are shown in Table 1. The CMC-AC biopolymer electrolyte solution was casted into petri dishes and left for a period of time (1 month) at room temperature to dry into thin film. Three (3) sample replicates were produced for each composition. The CMC-AC biopolymer films obtained are free standing with no phase separation observed (Appendix A).
(1)$$\mathrm{composition}=\;\frac{x}{x+y}\times 100$$

### 2.2. Electrical Impedance Spectroscopy (EIS)

A HIOKI 3532-50 LCR Hi-Tester (HIOKI, Nagano-ken, Japan) used to investigate the conductivity of CMC-AC biopolymer film. Each samples tested in the frequency range of 50 Hz to 1 MHz by utilizing stainless steel as blocking electrodes at a temperature range of 303–363 K.

### 2.3. X-ray Diffractometer (XRD)

The degree of crystallinity of CMC-AC biopolymer films obtained using Rigaku Miniflex II X-ray diffractometer (Rigaku, Tokyo, Japan) with *CuKα* radiation sources. The radiation sources directed the X-ray beam onto the films and scanned at angle, 2*θ* between 5*°* and 80*°*.

### 2.4. Fourier-Transform Infrared (FTIR)

Complexation or interaction between CMC and AC salts were determined by using a Thermo Nicolet 380 FTIR spectroscopy (Thermo Fischer, Madison, WI, USA) equipped with attenuated total reflection (ATR). The spectrum recorded in a wavenumber ranged from 800–4000 cm^−1^ and with resolution of 4 cm^−1^.

### 2.5. Transference Number Measurement (TNM)

The dc polarization techniques with 1.5 V fixed voltage were applied across the samples held by a stainless steel holder. A digital multimeter and computer were also connected to the circuit to collect the current (*I*) value against time (*t*). The transference number, *t_ion_* of CMC-AC biopolymer films was calculated from the testing.

## 3. Results

### 3.1. XRD Analysis

Figure 1a shows the X-ray diffraction (XRD) result of pure AC and CMC-AC biopolymer films. From the figure, pure AC salt displays several sharp peaks which represent the crystalline nature of AC salt with three prominent peaks can be observed at angle 2*θ* = 22.54°, 30.2° and 34.54°. The XRD pattern for CMC-AC biopolymer films show a similar pattern after the addition of AC salts where the amorphous hump is centered at 2*θ* = 21.1° (Figure 1a). It can be noticed that none of crystalline peak observed from AC salt appeared for all films, which indicates the AC salts dissolved within CMC amorphous phase to form an electrolyte system with improved electrolyte properties [26]. The amorphous phase will lead to higher ionic conductivity due to greater ionic diffusion, since the amorphous polymers have a flexible polymeric backbone [27]. Several reports have proven that the high amorphous phase has led to ionic conductivity improvement [28,29,30].

De-convolution techniques on XRD spectra can reveal the explicit details of the amorphous or crystallinity phase of electrolyte films [31]. The de-convoluted XRD diffractograms show several peaks where a narrow and sharper peak (solid green line) represents the crystalline phase and a broader peak (dotted blue line) represents the amorphous phase (Figure 1b). From the de-convoluted peak, the degrees of crystallinity (*X_c_*) of each CMC-AC biopolymer films were calculated using Equation (2) and the results are given in Table 2.
(2)$$\mathrm{degree}\;\mathrm{of}\;\mathrm{crystallinity},\;X_{c}\left(\%\right)=\frac{A_{c}}{A_{c}+\;A_{a}}\;\times \;100\%$$

The overlapping peaks and area forms under the diffractograms were determined for either amorphous (*A_a_*) or crystalline (*A_c_*) phases before calculating the *X_c_*, and the results are tabulated in Table 2. The degree of crystallinity, *X_c_* for the CMC-AC biopolymer films is in between 46% to 30% as shown in the table. The irregularity (random arrangement) of the polymer structure due to the increase in amorphous phase can led to lower energy barrier between hopping sites. Thus, the increasing amorphous phase allows for easier ion conduction which in this case is the H^+^ ion through the polymer matrix and subsequently increases the ionic conductivity value. The degree of crystallinity, *X_c_* trend of current work is similar compared to other works which the highest conducting sample has the lowest *X_c_* [32,33].

### 3.2. FTIR Analysis

Figure 2 shows the FTIR spectrum of the CMC-AC biopolymer films in the range of (a) 1500–1800 cm^−1^ and (b) 900–1200 cm^−1^. Overall the FTIR spectrum can be seen in Appendix A in the Appendix A. These ranges are responsible for the vibration of carboxyl group (COO) of CMC, where it is the probable site for interaction between CMC and AC [17,19,34]. The FTIR peak centered at 1056 cm^−1^ and 1591 cm^−1^ correspond to the vibration peak of C–O and C=O of the COO, respectively. As seen in Figure 2, there is no shifting of the peak indicating that the AC did not directly interact with that particular site. However, the C=O peak shows an observable reduction in intensity as the AC composition increases which implies that the AC salt weakly interacts with the COO. The presence of free lone pair electron at C=O attract free ions to interact. The small bump appeared around ~1650 cm^−1^ is believed due to the AC salt. The AC salt believed to dissociate to NH_4_^+^ and CO_3_^2−^ ions. This allows a protonation process between the loosely bonded H^+^ from NH_4_^+^ structure to the complex site [19]. The protonated H^+^ then able to transport to the conduction site (C=O) and subsequently improve the ionic conductivity of the electrolyte films. This can be further clarify from the impedance and transport analysis.

### 3.3. Ionic Conductivity and Transport Properties

The real (*Z_r_*) and imaginary (*Z_i_*) impedance data obtained from EIS testing presented in the form of Cole-Cole plot (Figure 3a,b). This plot reveals semi-circle at high frequency (left side of the plot) and slanted line at low frequency (right side of the plot). This shape can be explained in terms of the electronics component where the semi-circle represents parallel combination of capacitor and resistor [35]. The resistor represents the ionic migration through the polymer matrix while the capacitor represents the polarized polymer chains [35]. The slanted line is due to the electrode polarization effect where electrical double layer capacitance occur due to the blocking stainless steel electrode used in this testing [36]. The bulk resistance, *R_b_* value for each CMC-AC biopolymer film was obtained from the intersection of the semi-circle and slanted impedance line value at the *x*-axis [15]. The value of ionic conductivity is calculated using Equation (3). The *t* and *A* in the equation correspond to film thickness and the contact area, respectively. The ionic conductivity value at room temperature (303 K) against AC composition (wt.%) is plotted in Figure 3c. As seen in the figure, the ionic conductivity increases for sample AC1 (1 wt.%) to AC7 (7 wt.%) with the value of 7.71 × 10^−6^ Scm^−1^ and later dropped for sample AC9 (9 wt.%) and sample AC11 (11 wt.%). This was attributed to the diminishing effect of bulk resistance, which was apparent with the size changes to the impedance semi-circle as seen in Figure 3a,b, that indicates lower resistance for the mobile ions to move [17]. Table 3 shows the comparison of ionic conductivity of current work with other polymer-doped ammonium salt electrolyte systems. This work exhibits higher ionic conductivity when compared to other single ammonium and di-ammonium salt system. Figure 3d shows the ionic conductivity at elevated temperature for selected biopolymer films. In the investigated temperature range, the ionic conductivity increases as temperature increases. This is due to two reasons, (1) the AC absorbs the thermal energy and allows more ions dissociation, (2) the CMC backbone vibrate and create space for ionic conduction thus increase ionic conductivity at high temperature. This indicates the CMC-AC biopolymer films is thermally assisted which is proven from good fitting (R^2^ = 0.93–0.98) of the ionic conductivity to the Arrhenius relationship (Equation (4)). Transport properties analysis can further reveal the details behind the ionic conductivity improvement of the CMC-AC biopolymer system.
(3)$$\mathrm{Ionic}\;\mathrm{conductivity},\;\sigma =\frac{t}{R_{b}\;\times \;A}\;$$
(4)$$\sigma =\sigma_{o}exp^{\left(\frac{-E_{a}}{kT}\right)}$$

Arof et al. successfully calculated the transport number of their electrolyte system through FTIR peak (de-convoluted technique) [42]. Thus, the same technique was applied to the current work. The de-convoluted FTIR spectra were assigned into free ions and contact/aggregates ions to determine the area of de-convoluted peak before calculating the transport number of the electrolyte system. The FTIR spectra range of 1500–1750 cm^−1^ was selected for de-convoluted process to find the peak originated from ammonium (NH_4_^+^) ion. Figure 4a shows the de-convoluted peak of each CMC-AC biopolymer film. The peak centered at ~1589 cm^−1^ is the peak of the carboxylic group (COO–) of CMC [17]. The free ions peak centered from 1594–1643 cm^−1^ while the contact ion centered from 1650–1684 cm^−1^. The percentage of free ions were first calculated using Equation (5), where *A_f_* is the area of free ions peak and *A_c_* is the area of contact ions peak. Afterward, the percentage of free ions obtained were used to determine the transport number (number of mobile ions, *η* ionic mobility, *μ* and diffusion coefficient, *D*) using Equations (6)–(8) [43].
(5)$$\mathrm{Percentage}\;\mathrm{of}\;\mathrm{free}\;\mathrm{ions}\;\left(\%\right)=\;\frac{A_{f}}{A_{f}+A_{c}}\times 100\%$$
(6)$$\eta =\;\frac{M\;\times N_{A}}{V_{Total}}\times \%\;\mathrm{of}\;\mathrm{free}\;\mathrm{ions}$$
(7)$$\mu =\frac{\sigma }{\eta e}$$
(8)$$D=\frac{\mu k_{B}T}{e}$$

In Equation (6), *M*, *N_A_*, and *V_Total_* stands for the number of moles of salts used, Avogadro’s number and the total volume of electrolytes films respectively, where *V_Tota_*_l_ is equal to mass, *m* divided by density, *ρ* of each material used. In Equation (7), *σ* is the ionic conductivity of the electrolyte films and *e* is the electric charge. In Equation (8), *k_B_* is the Boltzmann constant and *T* is the absolute temperature in kelvin.

The peak for free ions originated from NH_4_^+^ ions, which have deprotonated into NH_3_^+^ ions. The peak position found from the de-convoluted plot is in good agreement with other reports [44,45]. The percentage of free ions peak is calculated and tabulated in Table 4. From the table, the CMC-AC biopolymer films has a percentage of free ions is in the range between 59% and 85%. For sample AC1–AC7, the percentage of free ions decreases from 85.48% for sample AC1 to 59.53% for sample AC7 which corresponds to the highest conducting sample, before it starts to increase for sample AC9 and AC11. This is in inverse manner compared to other reports where the reports show that the percentage of free ions is usually with the highest ionic conductivity sample [46,47,48]. However, the percentage value only gives a general view of the transport number. Further analysis of the transport number (Equations (6)–(8)) will further help to explain the transport behavior.

Figure 4b presents the calculated transport number. From the figure, the value of *η* increase with each AC composition added for sample AC1 to AC7 (low AC composition film). At low composition, AC salt is able to dissociate into its respective ions (NH_4_^+^ and CO_3_^−^) and has a small probability to form an ion cluster (contact ions), thus allowing one of the protons (H^+^) which is loosely bound to the NH_4_^+^ structure to be protonated and transported across the medium. As a result, the *μ* and *D* value is increased. At higher AC composition films (AC9-AC11), the value of *η* continues to increase in contrast to the ionic conductivity, which dropped along with *μ* and *D* value. The ions packed closely in the electrolyte when a huge amount ionic dopant supplied into the system. Consequently, this makes the de-protonation and protonation process at the ammonium (NH_4_^+^) ions occurs almost spontaneously and reduces the effective mobility of mobile ions, and consequently the diffusion rate thus prove *μ* and *D* dependency [47]. The dependency of *μ* and *D* towards ionic conductivity value is similar to the factors affecting ionic conductivity in a machine-learning framework proposed by Shi et al. [1], which mentioned diffusion coefficient and ionic radius as some of the factors. In this work, the diffusion coefficient improvement is believed due to small H^+^ ionic radii. Scheme 1 illustrates the behavior of ionic transport in the CMC-AC biopolymer at (a) low composition and (b) at high composition.

### 3.4. Transference Number Analysis

In this analysis, the biopolymer films were subjected to polarization currents (dc), which the current will become saturated indicating ionic transference, *t_ion_* value [48]. See Appendix A for testing setup. Figure 5 shows the plot of normalized current against time for sample AC7. From the figure, the initial total current decreases against time and eventually reached a steady state current. The value of *t_ion_* for each sample was normalized before calculated using Equation (9) and the corresponding value is tabulated in Table 4.
(9)$$t_{ion}=\frac{\left(I_{initial}-I_{steady-state}\right)}{I_{initial}}$$

The current become saturated due to ion accumulation at the blocking electrodes (increase resistance) while the remaining current flow is due to electron conduction. From the table, the sample with the highest ionic conductivity value (AC7) has the highest *t_ion_* (0.98). This verified that the optimal CMC-AC biopolymer film system (AC7) is primarily ionic which is desirable for an intercalation process at the electrode [1]. However, the ionic conduction in the electrolyte system can be either due to cations or anions. The number of mobile ions, *η* calculated previously is actually the overall number for both ionic species and ionic mobility, *µ* and diffusion coefficient, *D* is the mean value for cations and anions [42]. Thus it can be differentiated to cationic and anionic mobility (*µ_+_ µ_−_*) and cationic and anionic diffusion coefficient (*D_+_*, *D_−_*) and calculated using Equations (11) and (13), respectively [42,43]. Table 5 shows the results of the respective calculated cationic and anionic value obtained for CMC-AC biopolymer system where the value of *µ_+_* and *D_+_* is higher than *µ_−_* and *D_−_*. This indicates that in the current work the primary conducting species is cations, which further proved that the CMC-AC biopolymer film is also a proton (H^+^) conductor. The results obtained in this study is similar to the previous electrolyte system reported in the literature using another type of ammonium salt [49,50].
(10)$$D=\;D_{+}+D_{-}=\;\frac{k_{B}T\sigma }{\eta e^{2}}$$
(11)$$t_{ion}=\;\frac{D_{+}}{D_{+}+D_{-}}$$
(12)$$\mu =\;\mu_{+}+\mu_{-}=\;\frac{\sigma }{\eta e}$$
(13)$$t_{ion}=\;\frac{\mu_{+}}{\mu_{+}+\mu_{-}}$$

## 4. Conclusions

To conclude, the CMC-AC biopolymer film has been successfully prepared. Both CMC and AC salts dissolved completely and had low degree of crystallinity, *X_c_* for the highest conducting sample (AC7) as proven by the XRD de-convolution result, while the AC salt appeared to have weak interaction with carboxyl group of CMC. The highest ionic conductivity value for the system was 7.71 × 10^−6^ Scm^−1^ and thermally assisted (Arrhenius relations). The ionic conductivity of the current work was higher than other diammonium salts (ammonium adipate and ammonium sulfate) as presented in this work. It was also noted that the best composition (wt.%) of diammonium salt in a polymer electrolyte was lower compared to a single ammonium salt which was due to abundance of ammonium ions in a mol of ammonium salts. The transport number analysis showed that the CMC-AC biopolymer films were influenced by the ionic mobility, *μ,* and the diffusion coefficient, *D,* since both values achieved the maximum value for sample AC7, while the increased value of *η* for higher salt composition films led to lower *µ* and *D* values. The transference number also verified that the conducting species was mainly cation, which in this system was the proton (H^+^). The H^+^ was able to transport across the polymer matrix through the high amorphous phase of the electrolyte films and subsequently increased the ionic conductivity. This shows a promising prospect for CMC-AC biopolymer film in electrochemical applications. However, further enhancement is still needed to increase the ionic conductivity for suitable electrochemical applications (~10^−4^ Scm^−1^).

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
