# Peer review of "Natural Inspired Carboxymethyl Cellulose (CMC) Doped with Ammonium Carbonate (AC) as Biopolymer Electrolyte"

_polymers, 2020, doi:10.3390/polym12112487_

Round 1
Reviewer 1 Report
The manuscript polymers-900012 presents a study on Natural inspired carboxymethyl cellulose (CMC) doped with ammonium carbonate (AC) as biopolymer electrolyte. This work is in favor of the development of solid polymer electrolyte from biopolymer. However, some issues should be addressed before publication.
1. The authors provide the degree of crystallinity, for the CMC-AC biopolymer films is in between 4.3% to 9.1%. The Xc of AC7 is lowest among them and almost the same of that of AC0, but the Xc of AC9 is much higher than that of AC7 and AC11. the authors should give a reasonable explanation for this data.
2. To deepen the understanding of results and enrich the readability of the manuscript, it is necessary to cite and review the most relevant reference in a reasonable fashion which introduces the multi-scale computation simulation on modeling energy materials, e.g., Chinese Physics B 25(1), 018212 (2016)
3. The authors deconvolute the peak position from the XRD and FTIR spectra, the relative standard deviation or the residuals of a fit should be show.
4. The authors describe the impedance data obtained from EIS testing presented in the form of Cole-Cole plot, how do they confirm it.
5. The authors should provide to further test the great miscibility between the CMC functional group and AC salts by SEM.
Author Response
- The authors provide the degree of crystallinity, for the CMC-AC biopolymer films is in between 4.3% to 9.1%. The Xc of AC7 is lowest among them and almost the same of that of AC0, but the Xc of AC9 is much higher than that of AC7 and AC11. the authors should give a reasonable explanation for this data.
- The results have been re-analysed. New result is shown in table 2 (Line 144). Re-analysed result better represent the Xc to ionic conductivity.
2. To deepen the understanding of results and enrich the readability of the manuscript, it is necessary to cite and review the most relevant reference in a reasonable fashion which introduces the multi-scale computation simulation on modeling energy materials, e.g., Chinese Physics B 25(1), 018212 (2016)
- reference has been cited in line 232-233 linking ionic radius and diffusion coefficient to ionic conductivity improvement.
3. The authors deconvolute the peak position from the XRD and FTIR spectra, the relative standard deviation or the residuals of a fit should be show.
- The regression of the fit line and residual sum of square has been added and presented in table 2 (line 144) and table 3 (line 195).
4. The authors describe the impedance data obtained from EIS testing presented in the form of Cole-Cole plot, how do they confirm it.
- relevance explanation and citation (line166-173) has been added in section 3.3 to explain the cole-cole plot.
5. The authors should provide to further test the great miscibility between the CMC functional group and AC salts by SEM.
- The main objective of this research is to obtain polymer electrolyte with ionic conductivity at least ~10-4 Scm-1 which is the minimum value solid electrolyte before applying to electrochemical application. Since the conductivity for this system is two magnitude order to achieve this number, we decided to only do XRD testing to at least confirm that the there is no salt crystalline phase presence in the polymer. Amendments has been made in the statement line 117-119. We appreciate the suggestion for SEM testing and we will added SEM testing in future study.
Reviewer 2 Report
This manuscript demonstrates an easy-fabricated method to develop a solid polymer electrolyte, which is remarkably interesting concerning about the natural inspiration by biopolymers. After careful reviewed the manuscript, I only have one suggestion for the authors: change the Figure S1 to main text and I would like to see more details in this Figure about the biopolymers, such as the morphological characterization of the surface by atomic force microscopy (AFM) or scanning electron microscopy (SEM), in order to obtain the rugosity and/or homogeneity information about the films. Another question is regarding of the sample preparation, Why the time is too long (1 month)? Can you use temperature to accelerate the film preparation?
Author Response
This manuscript demonstrates an easy-fabricated method to develop a solid polymer electrolyte, which is remarkably interesting concerning about the natural inspiration by biopolymers. After careful reviewed the manuscript, I only have one suggestion for the authors: change the Figure S1 to main text and I would like to see more details in this Figure about the biopolymers, such as the morphological characterization of the surface by atomic force microscopy (AFM) or scanning electron microscopy (SEM), in order to obtain the rugosity and/or homogeneity information about the films.
- We appreciate the suggestion for SEM testing. However, this study is focused on the ionic conductivity improvement and ionic transport behavior with salt concentration. Thus, we would like to not added this testing and analysis in this work. However, we will highly consider this testing in the future.
Another question is regarding of the sample preparation, Why the time is too long (1 month)? Can you use temperature to accelerate the film preparation?
- The long time taken is because the sample was let to dry in room temperature (between 25 - 30 degree C). With regards to applying heat, we have tried to apply 50 and 60 degree in oven. But the ionic conductivity obtain is lower compared to the method used in the menuscript.
Reviewer 3 Report
The current work by Sohaimy et al. reports on doped carboxymethyl cellulose with ammonium carbonate for biopolymer electrolytes. A detailed investigation was performed including crystallinity, ionic conductivity, transference number, etc., by the authors. Importantly, the authors were well experienced in the field of biopolymer for energy applications. However, it is important to understand the novelty of this work by substituting the ammonium carbonate with ammonium nitrate and ammonium fluoride. The author needs to address some of the content to better convey the results properly.
1) The author needs to compare the results of ionic conductivity and other properties with existing ammonium salts to understand the enhancement of their properties by the introduction of one extra ammonium group in CMC.
2) Most of the polar polymers used in the preparation of polymer electrolytes are semicrystalline polymers, i.e., containing both crystalline and amorphous phases. Even after introducing 11 wt % of ammonium carbonate (AC) into the biopolymer, the crystallinity of AC is absent in XRD spectra. However, the authors suggest the miscibility of salt in the polymer. Did the author consider studying the effect of ammonium salts (AC, AN, AF) using DMA and TGA for glass transition, viscoelastic, and thermal properties?
3) How stable these biopolymer electrolytes?
4) The author needs to consider in measuring the dc conductivity at high temperatures as the salt concentration significant change in the conductivity behavior
5) Please comment on other ammonium salts such as (NH4)2SO4, (NH4)2S, etc. in the conclusion section.
6) Please verify the wt% of the electrolyte composition in table 1.
7) Please check the equation numbers on page 7 from 190 to 194.
8) Importantly, the authors need to show the reproducibility of all the measurements.
Author Response
1) The author needs to compare the results of ionic conductivity and other properties with existing ammonium salts to understand the enhancement of their properties by the introduction of one extra ammonium group in CMC.
- The comparison with other ammonium salt polmer electrolyte systems have been added in section 3.3 with table 3 (180-195) for visualization.
2) Most of the polar polymers used in the preparation of polymer electrolytes are semicrystalline polymers, i.e., containing both crystalline and amorphous phases. Even after introducing 11 wt % of ammonium carbonate (AC) into the biopolymer, the crystallinity of AC is absent in XRD spectra. However, the authors suggest the miscibility of salt in the polymer. Did the author consider studying the effect of ammonium salts (AC, AN, AF) using DMA and TGA for glass transition, viscoelastic, and thermal properties?
- We appreciate the testing suggested from reviewer. Our main objective for this research is to first determine a biopoylmer based electrolyte system with high ionic conductivity at room temperature. Once achieved, we will look into those properties as suggested. At the moment we only look at the crystalline phase and the transport behavior when doped with ammonium carbonate to at least confirm the prospect of this new biopolymer films.
3) How stable these biopolymer electrolytes?
- The physical physical stability for this biopolymer electrolyte system has been mentioned Line 89-90.
4) The author needs to consider in measuring the dc conductivity at high temperatures as the salt concentration significant change in the conductivity behavior
- The ionic conductivity at high temperature has been added and explained in first paragraph of section 3.3, line 173 – 180.
5) Please comment on other ammonium salts such as (NH4)2SO4, (NH4)2S, etc. in the conclusion section.
- Comments on other ammonium salts has been added in the conclusion section (line 273-277).
6) Please verify the wt% of the electrolyte composition in table 1.
- Equation 1 (line 91) has been added in the manuscript to show the calculation for weight composition obtained in this study.
7) Please check the equation numbers on page 7 from 190 to 194.
- Equation numbering has been corrected on the equation and also in the paragraph explaining the equation (line 200 – 204).
8) Importantly, the authors need to show the reproducibility of all the measurements.
- The regression of the fit line and residual sum of square has been added and presented in table 2 (line 144) and table 3 (line 195).
Round 2
Reviewer 1 Report
Authors have revised the manuscript point by point. I recommend that the present version should be accepted for publication.
Reviewer 3 Report
Please accept in the present form